# Processing of Emotional Words by Czech-German Bilinguals

Vojtěch Kocourek, supervisor: Nikola Paillereau

## Introduction

The thesis examined the differences in word processing and evaluation between monolingual German speakers and bilingual Czech-German speakers with respect to emotional dimensions. The research questions were formulated on the basis of three concepts: a) The dimensional approach to emotionality, which does not see emotional words as a separate category, but rather assumes that the two dimensions used for their description – valence (positive/negative) and arousal (low/high level of activation) – represent a universal aspect of all words (Hinojosa et al., 2019). b) The effect of reduced emotional resonance in L2 which describes the tendency of speakers to perceive languages which they learned later in life or less well as less emotional (Harris et al., 2003). c) The hypothesis of emotional context of learning, which suggests that the level of emotional resonance of words depends on whether they were learned and used in emotional contexts (Caldwell-Harris, 2014).

## Method

An experiment was conducted with 16 German monolinguals and 19 Czech-German bilinguals. After a training phase, all participants were asked to rate an auditorily presented word by clicking on a two-dimensional emotional scale (see Figure 1), while at the same time they were being monitored by fNIRS for differences in brain activity in relevant cortical areas. The aim was to obtain behavioural and neurophysiological data corresponding to each of the 109 German words from both groups. Based on the hypotheses about emotional resonance and context of learning, it was expected that either there would be no difference between the groups, or the responses of the bilingual group would indicate lesser emotionality, depending on the context in which they learned German.

## Results

Due to a technical error, the neurophysiological data could not be used, thus only the behavioural data were analysed. The measured ratings of valence did not significantly differ between the groups, while the ratings of arousal were significantly higher in the bilingual group (see Figure 2). The bilinguals did not differ among themselves in the levels of arousal as a function of their age of learning German, which was chosen as a stand-in value representing the context of acquisition. The number of respondents in combination with the character of distribution of their results did not allow for further statistical analysis.

## Discussion

The similarity of valence ratings is consistent with the hypotheses, suggesting that the bilingual speakers' acquisition of German happened in a sufficiently emotional context. However, the arousal ratings could not be accounted for by the selected concepts and after considering multiple perspectives, the higher levels among bilinguals were interpreted as a result of perceived situational pressure. No effect of reduced emotional resonance was observed. As the behavioural results could not be verified with complementary neurophysiological data and the number of participants was restrained, the generalisability of the findings is limited.

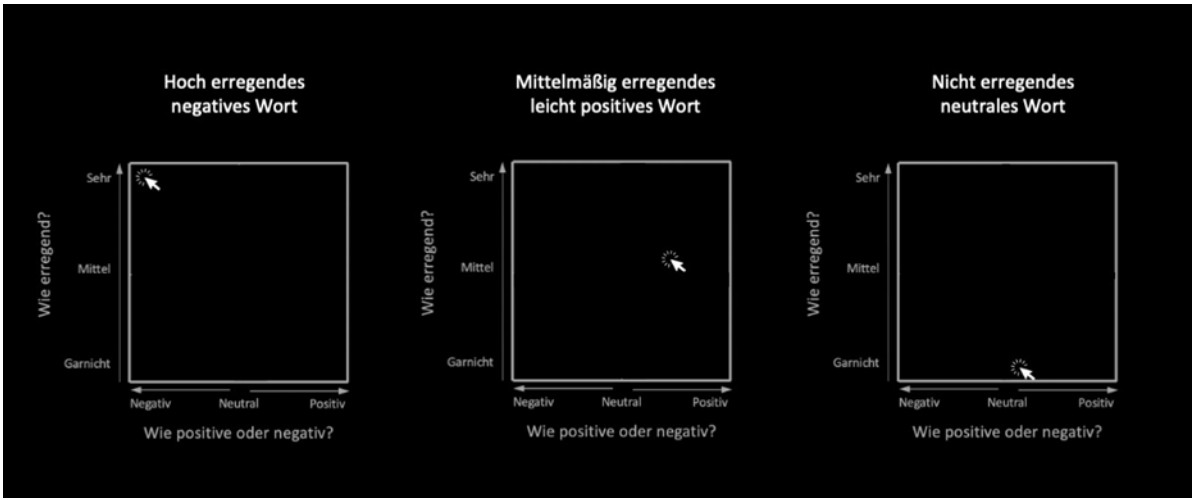

*Figure 1 Display from the training phase, examples of a word with high arousal and negative valence (left), medium arousal and slightly positive valence (middle) and low arousal and neutral valence (right)*

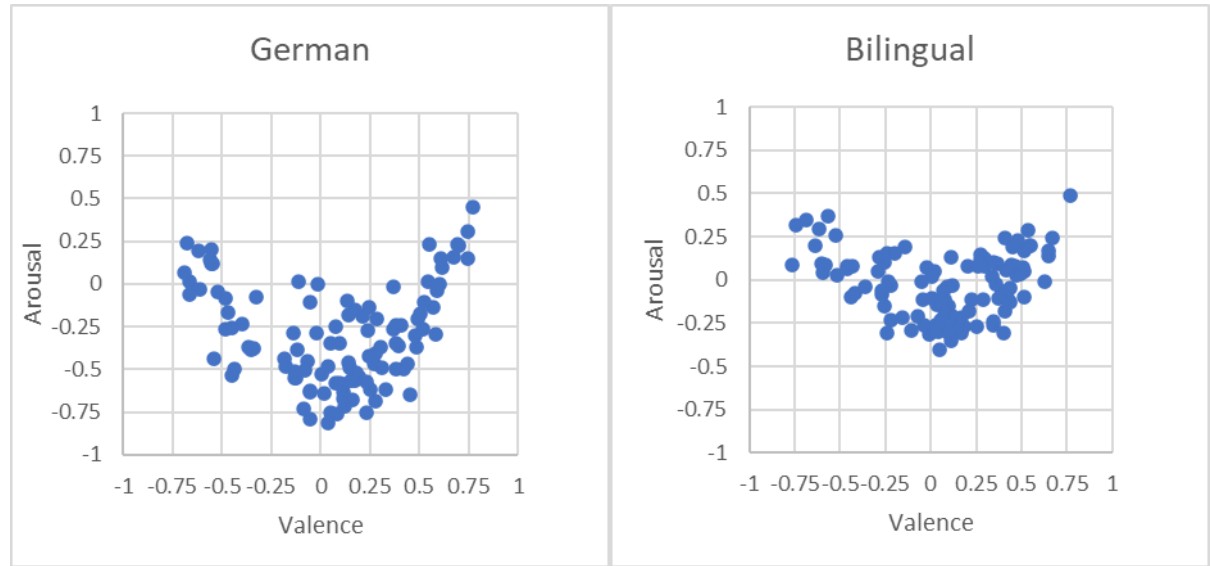

*Figure 2 Comparison of average group ratings of individual words on scales from -1 to 1 illustrating the difference in average arousal.*

**Key words:** *bilingualism, emotionality, arousal, valence, context of learning*

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
