# OpenReview forum: "Processing of Emotional Words by Czech-German Bilinguals"
_CUNI.cz/2024/CJOLPhD — CUNI 2024 CJOLPhD Submission_

### Official Review · ~Maria_Onoeva1 · 2025-01-06
**Good job!**

A very curious design of the experiment, I'm sorry there was a technical issue, it would be great to look at that type of data. The research questions are clearly stated with respect to the previous work and literature. The results are particularly interesting in relation to the hypotheses, although more research is of course needed. I also appreciate how well it is written and structured, especially given the limited space.

I have the following questions:
1) I didn't quite get what types of words were tested, was it a special category ("emotional words"?) or just any word? How were they picked for the experiment? It is useful to provide a couple of examples or a link to a repository with a table containing average ratings.
2) Another question is about age of participants, were they roughly the same age? Could also gender influence the results?
3) Does it make sense to test Czech speakers for similar words in Czech and then compare their reactions to German words?

Typographical suggestions:
- Bold face section names on one line with paragraphs for more space

---

### Official Review · ~Lucie_Jarůšková1 · 2025-01-06
**Great abstract**

The topic of the abstract is an interesting one, studied in modern psycholinguistics, and contributes to a better understanding of bilingualism as such and the perception of emotional words.

It is written in good professional English.

The structure of the abstract is clear, partly thanks to the use of headings. I recommend saving space and placing the headings directly next to the text (no line breaks). The abstract presents the subject matter and the three main theoretical points well. There is a slight lack of information about how the words were selected (the preparation of stimuli), although it is understandable that space is limited.

The statistical model used to analyze and compare the groups could be briefly mentioned in the results, which would help to present the results better. This is not an *error*, but a recommendation for improvement and better "selling" of the results.

My questions:
1.  It would also have been useful to indicate whether bilingual participants differed in their level of German proficiency (as the abstract already presents the results; it is not automatic that all bilingual speakers reach C2 in all domains). The text shows that the age of acquisition was taken as a factor. Does this mean that their level of German was same or different?
2. And, if / how was the *"the context in which they learned German"* and especially emotional words examined more? Were they dominant in Czech or German?

I appreciate the addition of a graph of the results and a screenshot illustrating the design of the experiment itself. The inclusion of references and mention of the limitations of the study are also great.

Overall, this is a very good abstract that presents an interesting field and experiment. Looking forward to more studies and results!

---

### Official Review · ~Radek_Šimík1 · 2025-01-08
**Very nice abstract with clearly stated objectives, method and results**

I don't have many comments on form. The abstract is clearly and correctly structured and clearly written. Maybe just one minor point: better quality of figures would be appreciated (ideally in vector format, svg, or embedded pdf).

Some comments on substance:

- The participant sample is well-characterized, but we learn little from the abstract about the words that were evaluated. How were they chosen? Did they fall into different categories? If yes, it would have been good to have the different categories visualized separately (e.g. by different colors of datapoints in the plots).
- I'm not sure it's possible to draw the inference from this work that the "acquisition happened in a sufficiently emotional context". That can be inferred only if it is really the case that emotional resonance is the result of "emotional learning". But this is not stated as a fact in the introduction, but rather just a hypothesis. So either the hypothesis si not a hypothesis, but a well-supported fact, or the inference is conditional on the the hypothesis being borne out in the future.

---

### Official Review · ~Barbora_Genserová1 · 2025-01-08
**Great!**

Clearly structurred abstract written in high-level English. For more space (if needed), the section names could be in bold and not on a separate line and numbered references in text.
I would expect more information about the selection of participants and how (if) they differed in their level of German, and of the words used in the experiment.